# Learning from Distinction: Mitigating backdoors using a low-capacity model

## ABSTRACT

Deep neural networks (DNNs) are susceptible to backdoor attacks due to their black-box nature and lack of interpretability. Backdoor attacks intend to manipulate the model's prediction when hidden backdoors are activated by predefined triggers. Although considerable progress has been made in backdoor detection and removal at the model deployment stage, an effective defense against backdoor attacks during the training time is still under-explored. In this paper, we propose a novel training-time backdoor defense method called Learning from Distinction (LfD), allowing training a backdoor-free model on the backdoor-poisoned data. LfD uses a low-capacity model as a teacher to guide the learning of a backdoor-free student model via a dynamic weighting strategy. Extensive experiments on CIFAR-10, GTSRB and ImageNet-subset datasets show that LfD significantly reduces attack success rates to 0.67%, 6.14% and 1.42%, respectively, with minimal impact on clean accuracy (less than 1%, 3% and 1%).

## CCS CONCEPTS

• **Security and privacy** → **Malware and its mitigation**; • **Computing methodologies** → **Computer vision**.

## KEYWORDS

Backdoor attack, Neural Networks

## 1 INTRODUCTION

Deep neural networks (DNNs) have achieved unprecedented success due to their remarkable performance in many applications, such as image classification[41], object detection[63], and traffic accident detection[19, 22]. Its superiority relies on model training on large datasets using intensive computational resources[40]. In this process, if the data is obtained from third-party data sources or the model is trained on a third-party platform, a malicious attacker is able to insert predefined triggers into the data samples[1, 13, 38] or directly modify the model's parameters to implant a backdoor into the model[4, 10, 39]. In addition, using pre-trained models or outsourced Machine Learning as a Service (MLaaS) provides attackers the opportunities to backdoor the model, though they may offer substantial performance improvements at a low cost. As a result, the attacked models perform normally on clean data, whereas misclassifying specific data to the target class when predefined

Permission to make digital or hard copies of all or part of this work for personal or classroom use is granted without fee provided that copies are not made or distributed for profit or commercial advantage and that copies bear this notice and the full citation on the first page. Copyrights for components of this work owned by others than the author(s) must be honored. Abstracting with credit is permitted. To copy otherwise, or republish, to post on servers or to redistribute to lists, requires prior specific permission and/or a fee. Request permissions from permissions@acm.org.
*ACM MM, 2024, Melbourne, Australia*
© 2024 Copyright held by the owner/author(s). Publication rights licensed to ACM.
ACM ISBN 978-x-xxxx-xxxx-x/YY/MM
https://doi.org/10.1145/nnnnnnn.nnnnnnn

triggers are added, which poses a severe security threat to DNN applications[13, 49].

To alleviate this threat, existing defense methods can be categorized into two aspects: backdoor detection methods[11, 12, 49] and backdoor removal methods[3, 24, 54]. Backdoor detection methods aim to identify whether a given model contains a backdoor trigger or whether a given dataset contains poisoned data. Backdoor removal methods are designed to remove backdoors during or after the model training process while preserving the model performance. Although the after-training removal methods have shown promising defense results, their effectiveness relies on the availability of a benign dataset, rendering them unsuitable for scenarios where clean data is not accessible[33, 55]. Training-time removal methods aim to detect potential poisoned data during the training of the model and inhibit the model from learning backdoor features from these poisoned data so as to obtain a backdoor-free model[21, 27, 31]. Such methods implemented during training often involve utilizing additional datasets to identify poisoned data[15, 49] or training multiple models ensemble[12, 21, 27], which suffering from significant computational costs. Recently, Li et al. proposed a two-stage unlearning technique (ABL) that allows for training a clean model on a poisoned dataset[31]. However, this method presents challenges in terms of inaccurate data isolation and reduced accuracy of purified model. Currently, effective defense methods against backdoor attacks during the training phase remain a crucial problem and need further exploration.

To fulfill the requirement of effectively mitigating backdoors during training, we proposes a novel training-time defense method named Learning from Distinction (LfD), which allows training a backdoor-free model on backdoor-poisoned data. Specifically, LfD has two stages. In the first stage, it fine-tunes a low-capacity model on the backdoor-poisoned dataset for a few epochs and then utilizes this model as a teacher model to supervise the training of a backdoor-free student model. In the second stage, a dynamic weighting strategy is adopted during training to flexibly select the poisoned data from the poisoned datasets. We theoretically analyze the reason why low-capacity models can act as the teacher to help with backdoor defenses. Empirical results across various datasets and model architectures demonstrate that our proposed LfD method achieves superior performance compared to existing state-of-the-art defenses.

In summary, our contributions are as follows:

- We present the impact of model capacity on the distinction of the loss value between backdoored and clean data during training. Utilizing a lower capacity model makes the distinction of the loss value between backdoored and clean data more prominent, which can contribute to identifying the poisoned data.
- We propose a simple but effective training-time backdoor defense method named Learning from Distinction (LfD),

which allows training a clean student model on a backdoored dataset with utilizing a low-capacity model as a teacher model.

- We conduct extensive experiments on CIFAR-10, GTSRB and ImageNet-subset datasets to validate the effectiveness of LfD against 9 well-known backdoor attacks. The results show that LfD outperforms existing state-of-the-art defenses and significantly reduces attack success rate with minor impact on clean accuracy.

## 2 RELATED WORK

### 2.1 Backdoor Attacks

Existing backdoor attacks can be classified into two main categories: (1) **Dirty-label attacks** insert trigger patterns into data and change their labels to a specific target label, associating the trigger with the target label. Gu et al. first introduced backdoor attacks in deep learning[13]. Subsequently, several studies focused on making poisoned data indistinguishable from clean data to evade human inspection, some of them replace triggers with subtle perturbations[28, 37, 51, 62], and others disperse the trigger to a much larger area of the image[1, 36]. However, in the above attacks, the source labels of the poisoned data are still different from the target label, making them detectable by examining the image-label relationship. (2) **Clean-label attacks** only modify the data whose source labels are the same as the target label. Hence, they do not need to change the data's label, making them more covert compared to dirty-label attacks. Several studies focus on modifying data and adding easy-to-learn trigger patterns on them to perform clean-label attacks[1, 51]. Nevertheless, compared to dirty-label attacks, clean-label attacks often suffer from lower attack success rate[29, 59].

### 2.2 Backdoor defenses

Recently, numerous defense methods have been proposed to mitigate the threat of backdoor attacks. Existing defense methods can be broadly categorized into two categories based on when they take effect: (1) **The defense methods** adopted during the model deployment phase aim to detect whether the input data [2, 11, 16, 47, 49, 56] or the model [3, 24, 43, 53] has been backdoor-poisoned and to remove the existing backdoor in the backdoor-poisoned model [30, 55, 57]. The anomaly input detection methods employ the concept of outlier detection to filter out backdoor-poisoned data [9, 11, 20, 23, 46], while the backdoored model detection methods utilize meta-classifiers [18, 24, 56] or analyze the model's internal structure [34, 61] to identify the presence of backdoors. Unlike detection methods, removal methods' purpose is to eliminate the impact of backdoor attacks. Some removal methods reconstruct the trigger and employ fine-tuning to repair the model[5, 8, 14, 32, 48, 53]. Apart from those trigger-reconstruct defenses, other approaches have been widely applied in removal backdoors, e.g., pruning[33, 55, 60] and model distillation[30]. (2) **Training-time backdoor defense methods** intend to train backdoor-free models on backdoor-poisoned datasets. Some approaches achieve this by extracting features of clean data from additional datasets to identify the poisoned data in the training dataset[15, 49]. Other approaches, on the other hand, focus on training multiple model ensembles and

applying voting mechanisms to mitigate the impact of poisoned data[12, 21, 27]. Recently, Anti-backdoor learning (ABL) isolates a portion of poisoned data and unlearns these data in the last few epochs of training to eliminate the backdoor[31]. However, unlearning can lead to the loss of semantic features and a decrease in the accuracy of the model. Our proposed defense method LfD belongs to the training-time backdoor defense method.

## 3 BACKGROUND AND PRELIMINARY ANALYSIS

In this section, we first present the utilization of gradient ascent training for backdoor defense. Subsequently, we delineate our observations regarding distinct learning behaviors between the losses of poisoned and clean data . Finally, we propose a method to amplify this distinction.

### 3.1 Backdoor defense with gradient ascent

In the backdoor attack scenario, the goal of the adversary is to inject triggers into the model by solving an optimization problem as follows:

$$L(\theta_t) = \mathbb{E}_{(x,y)\sim D_{poisoned}}(\ell(f_{\theta_t}(x), y)) + \mathbb{E}_{(x,y)\sim D_{clean}}(\ell(f_{\theta_t}(x), y)),\tag{1}$$

where $t$ represents the number of training epochs, $D_{clean}$ and $D_{poison}$ represent the set of clean data and poisoned data, respectively, $D_{clean} \cup D_{poison} = D_{train}$, and $\ell(\cdot)$ denotes the loss function which computes the distance between the predicted label $f_\theta(x)$ and the ground truth label $y$. Then, the minimum value of the loss is calculated by using the gradient descent optimization method, thereby improving model's classification accuracy. This equation indicates that the entire learning task is decomposed into two sub-tasks: one is to minimize $f_\theta$ on clean data, and the other is to minimize $f_\theta$ on poisoned data. Therefore, the trained backdoored model exhibits high classification accuracy on clean data and high attack success rate on poisoned data.

To mitigate the impact of backdoor attacks on the model, the gradient ascent training method can be employed to alleviate the influence of poisoned data [31]. The process is as follows:

$$L(\theta_t) = \mathbb{E}_{(x,y)\sim D_{clean}}(\ell(f_{\theta_t}(x), y)) - \mathbb{E}_{(x,y)\sim D_{poisoned}}(\ell(f_{\theta_t}(x), y)),\tag{2}$$

As shown in the above equation, maximizing the loss of poisoned data can effectively amplify the distance between model predictions and their labels, i.e., target labels. This prevents the model from classifying poisoned data as the target label, thus reducing the model's attack success rate. Therefore, if we can devise a method for precisely identifying poisoned data within the dataset, we can integrate it with the gradient ascent training approach to mitigate the impact of backdoor attacks on the model.

### 3.2 Distinct Behaviors In Learning Between clean And Poisoned Data.

Previous studies have identified poisoned data within datasets through the analysis of data loss. This is owing to the nature of backdoor attacks, which require the establishment of explicit associations between triggers and target labels to ensure that poisoned

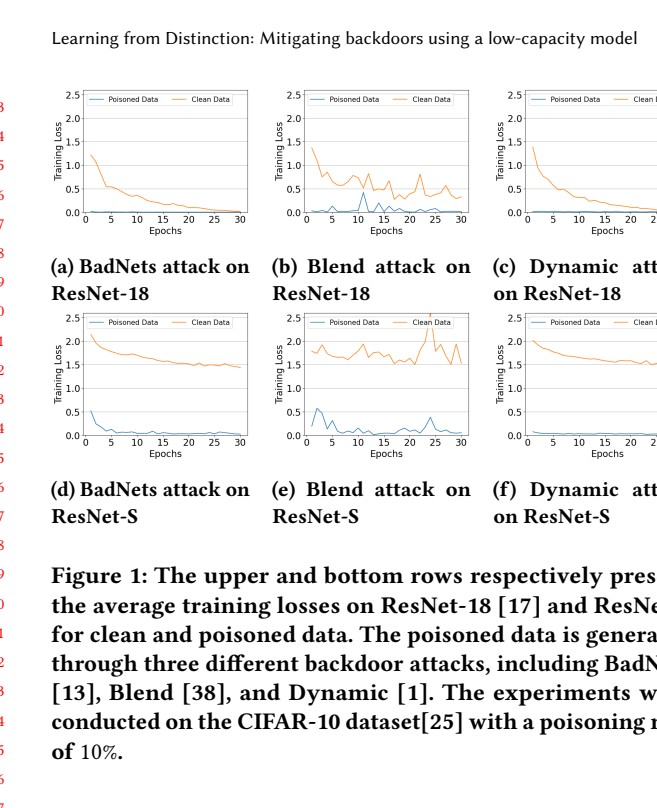

(a) BadNets attack on ResNet-18

(b) Blend attack on ResNet-18

(c) Dynamic attack on ResNet-18

(d) BadNets attack on ResNet-S

(e) Blend attack on ResNet-S

(f) Dynamic attack on ResNet-S

Figure 1: The upper and bottom rows respectively present the average training losses on ResNet-18 [17] and ResNet-S for clean and poisoned data. The poisoned data is generated through three different backdoor attacks, including BadNets [13], Blend [38], and Dynamic [1]. The experiments were conducted on the CIFAR-10 dataset[25] with a poisoning rate of 10%.

data is classified into the target label rather than its ground truth label. Consequently, poisoned data is found to be more easily learnable than clean data, resulting in lower losses[31]. To demonstrate this inference, we apply three classic backdoor attacks, named BadNets[13], Dynamic[38], and SIG[1] to construct three poisoned datasets with an injection rate of 10% on CIFAR-10 training data. Subsequently, we train the ResNet-18 model[17] on these poisoned datasets following the same configurations in Section 5. As shown in Figure 1a, 1b, 1c, for all three types of backdoor attacks, the average loss of clean data is higher than that of poisoned data, and this trend is more significant in the early stages of training. The above observations suggest that the loss of poisoned data is usually lower than that of clean data, especially in the early stage.

However, this method of identifying poisoned data has its shortcomings. Despite the significant difference in average losses between poisoned and clean data, there is still a possibility of poisoned data having losses similar to that of clean data. In Figure 2, we plot the number of poisoned data with losses greater than the average loss of clean data at the early training stage on the ResNet-18 model, and it is evident that such poisoned data exist in three poisoned datasets. Since some powerful backdoor attacks can succeed with just a tiny number of poisoned data[31], it is necessary to employ a strategy that amplifies the distinction in the losses of these two types of data in order to accurately identify the poisoned data in the dataset.

## 3.3 Impact Of Model Capacity On Data Loss

The capacity of a neural network model signifies the complexity of features it can learn. A model's capacity can be measured through its intricacy, such as the quantity of neurons of the model. Based on prior research, the generalization error of neural networks can

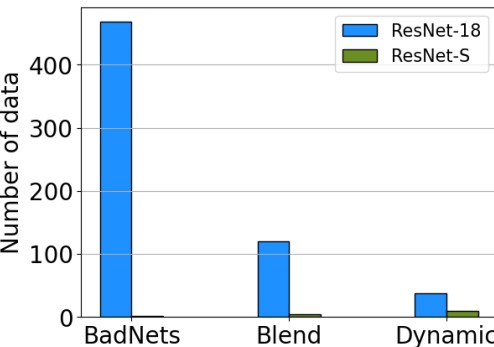

Figure 2: The number of poisoned data with losses higher than the average loss of clean data on ResNet-18 and ResNet-S at the fifth epoch.

be approximately formalized to a function of the model and dataset size [40]. Given a fixed dataset size, the following equation holds:

$$\hat{\epsilon}(m) = \epsilon_0 \left\| \frac{bm^{-\beta} + c_\infty}{bm^{-\beta} + c_\infty - i\eta} \right\|, \tag{3}$$

where $\hat{\epsilon}$ denotes the classification error on the test dataset, $m$ denotes the number of parameters in the model, $\beta(\beta \geq 0)$ control the global rate at which error decreases with the sizes of the model, $i = \sqrt{-1}$, and $c_\infty(c_\infty > 0)$ is the asymptotic lower value attainable. Here the simple pole at $\eta$ controls the transition point from the initial random-guess level $\epsilon_0$ as $(m)$ increase.

The above equation showed that the classification error increases as the number of model parameters decreases. This implies that the model's capability to accurately classify data decreases with a reduction in its capacity. Moreover, since the loss of data measures the classificatuon accurcy of the model, the data's losses on the model usually rise with a reduction in model capacity. Simultaneously, according to Section 3.2, poisoned data is more easily-learnable compared to clean data. Therefore, we claim that reducing the model's capacity could render it less adept at learning clean data while retaining its capability to learn poisoned data. This would effectively amplify the disparity in loss between the two types of data. To substantiate this point, we construct a ResNet-based model with fewer parameters than ResNet-18, referred to as ResNet-S, described in detail in Section 5.3, as a comparative model to ResNet-18 in order to observe how the difference in the average loss of clean and poisoned data during model training varies with the model capacity.

As depicted in Figure 1, across various backdoor attack scenarios, the average loss of clean data on ResNet-S is notably higher than that on ResNet-18. However, the average loss of poisoned data exhibits no significant disparity between these two models. The above findings validate our proposition that, despite the difficulty of ResNet-S in learning clean data due to its low capacity, it still has the ability to learn poisoned data, which leads to a more pronounced distinction in the average loss between the two types of data on ResNet-S.

Following the above results, we test the number of poisoned data with losses greater than the average loss of clean data on ResNet-18 and ResNet-S. We conducted experiments on ResNet-18 and ResNet-S to evaluate the quantity of poisoned data with losses greater than the average loss of clean data. As depicted in Figure 2, the number of such data instances on ResNet-S is notably reduced compared to ResNet-18, thereby further confirming the conclusions drawn in the preceding discussion.

Furthermore, we constructed low-capacity models on several other architectural frameworks for additional exploration. Specific results are presented in Section 5.3.

# 4 PROPOSED METHOD: LEARNING FROM DISTINCTION

Based on the observations and analyses in Section 3, we propose a Learning from Distinction (LfD) approach which involves training a clean model on the poisoned data set $D_{train}$ within the context of a supervised training process. As illustrated in Figure 3, we decompose the entire training process into two stages: training a teacher model $f_{\theta_{teacher}}$ capable of discerning the differences between poisoned and clean data, and using $f_{\theta_{teacher}}$ to dynamically discriminate poisoned data within $D_{train}$ to supervise the training of the clean model $f_{\theta_{student}}$.

**Stage 1: Training a $f_{\theta_{teacher}}$ capable of distinguishing between clean and poisoned data sets**.

First and foremost, we require a $f_{\theta_{teacher}}$ capable of distinguishing between clean and poisoned data within $D_{train}$. Drawing insights from the observations described in Section 3, it becomes evident that models with low capacity face challenges in classifying clean data. Nevertheless, these models still exhibit an intrinsic aptitude for adeptly classifying poisoned data, especially during the initial phases of training. This phenomenon engenders a noticeable discrepancy in losses between these two data categories. Consequently, we embark on creating a low-capacity model by reducing the number of neurons within the model, followed by subjecting it to a few epochs of training solely on $D_{train}$. This maneuver ensures a pronounced dissimilarity between poisoned and clean data on this model. Subsequently, this model assumes a pivotal role in the context of Learning from Distinction (LfD), functioning as the cornerstone of $f_{\theta_{teacher}}$.

**Stage 2: Utilizing $f_{\theta_{teacher}}$ for the dynamic discrimination of poisoned data to supervise the training process of a backdoor-free model $f_{\theta_{student}}$**.

Subsequently, we identify potential poisoned data within $D_{train}$ based on their losses on $f_{\theta_{teacher}}$, followed by gradient ascent training on these identified data to train a clean $f_{\theta_{student}}$ on $D_{train}$. This stage aims to mitigate the impact of poisoned data in the training process of $f_{\theta_{student}}$. To identify the poisoned data within $D_{train}$, we initially adaptively establish a threshold based on the losses of data in $D_{train}$ on $f_{\theta_{teacher}}$:

$$\gamma = DT(((f_{\theta_{teacher}}(x_1), y_1) \cdots (f_{\theta_{teacher}}(x_n), y_n)), index), \\ \text{s. t. } index = n \times \alpha, \tag{4}$$

where $n$ represents the number of data in $D_{train}$, and $\alpha$ is a hyperparameter, the $DT()$ function first calculates the losses of all data points in $D_{train}$ with respect to $f_{teacher}$, then sorts the losses

---

**Algorithm 1** LfD

1: Initialization: $f_{teacher}, f_{teacher}, D_{train}$
2: $\gamma = DT(((f_{teacher}(x_1), y_1) \cdots (f_{teacher}(x_n), y_n)), index);$
3: **for** each $data \in D_{train}$ **do**
4: $\quad w_{(x,y)} = \begin{cases} \frac{\ell(f_{teacher}(x), y)}{\max_{i=1}^n(\ell(f_{teacher}(x_i), y_i))}, & \ell(f_{teacher}(x), y) > \gamma \\ \frac{\beta \cdot (\ell(f_{teacher}(x), y) - \gamma)}{\max_{i=1}^n(\ell(f_{teacher}(x_i), y_i))}, & \ell(f_{teacher}(x), y) \le \gamma, \end{cases}$;
5: **end for**
6: When updating $f_{teacher} : L(f_{student}) = \mathbb{E}_{(x,y) \sim D_{train}}(w_{(x,y)} \cdot \ell(f_{student}(x), y)))$
**Output:** Backdoor-free model $f_{student}$

---

in ascending order, and selects the loss at the $index$-th position as the threshold. We perform gradient ascent training on the data in $D_{train}$ with losses below this threshold. Specifically, as shown in Equation 1, when updating $f_{\theta_{student}}$, we subtract the losses of these data points to decrease the probability of $f_{\theta_{student}}$ classifying them into their labels. Though we cannot guarantee that the loss of every poisoned data in $D_{train}$ is below the threshold, training a substantial portion of the poisoned data with gradient ascent can still mitigate the impact of these unrecognized poisoned data. Therefore, the loss function employed for updating $f_{\theta_{student}}$ is presented as follows:

$$L(\theta_{\text{student}}) = \mathbb{E}_{(x,y) \sim D_{\text{non-candidate}}} (\ell(f_{\theta_{\text{teacher}}}(x), y)) \\ - \mathbb{E}_{(x,y) \sim D_{\text{candidate}}} (\ell(f_{\theta_{\text{teacher}}}(x), y)), \\ \text{s.t. } (x, y) \in \begin{cases} D_{\text{non-candidate}}, & \text{if } \ell(f_{\theta_{\text{teacher}}}(x), y) > \gamma, \\ D_{\text{candidate}}, & \text{if } \ell(f_{\theta_{\text{teacher}}}(x), y) \le \gamma, \end{cases} \tag{5}$$

However, the gradient ascent training method may have an impact on the accuracy of $f_{\theta_{student}}$ on clean data as we presented in Section 5.2. We posit that this phenomenon arises due to the some clean data instances have losses below the threshold, and performing gradient ascent training on these data reduces the classification accuracy of $f_{\theta_{student}}$. Therefore, we introduce a weighting method that dynamically assigns weights to data to determine their significance during training, thereby mitigating the impact of gradient ascent training on the accuracy of $f_{\theta_{student}}$. The weight for each data in $D_{train}$ is computed as follows:

$$L(\theta_{student}) = \mathbb{E}_{(x,y) \sim D_{train}}(w_{(x,y)} \cdot \ell(f_{\theta_{teacher}}(x), y))), \\ \text{s. t. } w_{(x,y)} = \begin{cases} \frac{\ell(f_{\theta_{teacher}}(x), y)}{\max_{i=1}^n(\ell(f_{\theta_{teacher}}(x_i), y_i))}, & \ell(f_{\theta_{teacher}}(x), y) > \gamma \\ \frac{\beta \cdot (\ell(f_{\theta_{teacher}}(x), y) - \gamma)}{\max_{i=1}^n(\ell(f_{\theta_{teacher}}(x_i), y_i))}, & \ell(f_{\theta_{teacher}}(x), y) \le \gamma, \end{cases} \tag{6}$$

where the weight of data $(x, y)$ is denoted as $w_{(x,y)}$ in the equation, the hyperparameter $\beta$ determines the importance of gradient ascent training relative to gradient descent training. Different from equation 5, equation 6 calculates $L(\theta_t)$ by considering the weight of each data point based on its loss on $f_{\theta_{teacher}}$. Compared to the solely unweighted gradient ascent training, even when some clean data possess losses below the threshold and are assigned negative weights, their relatively lower losses compared to most poisoned

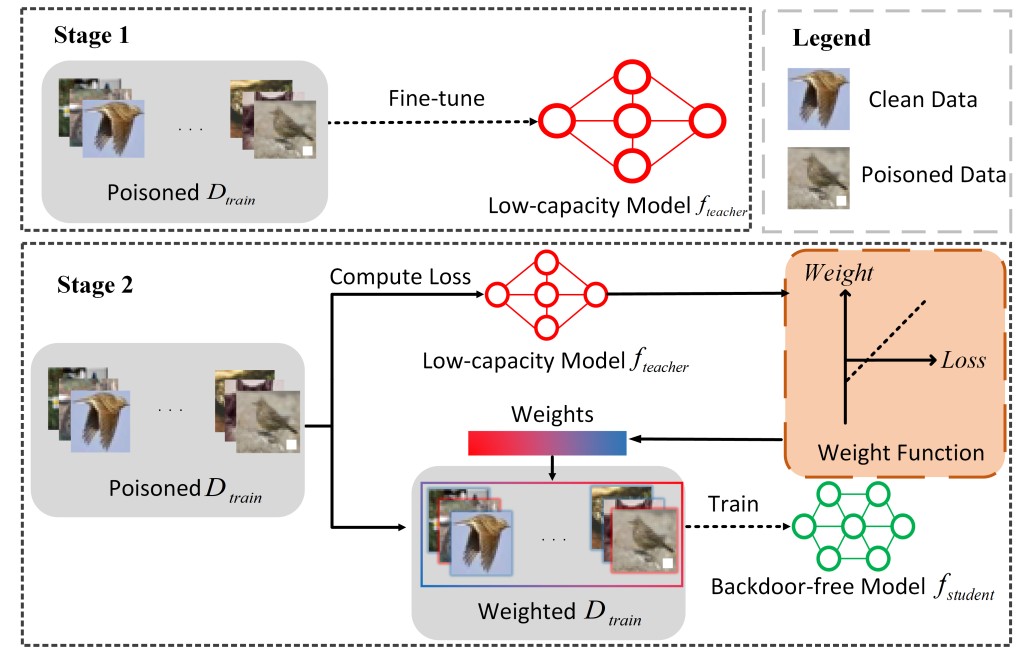

**Figure 3: Pipeline of our LfD. It consists of two main stages. (1) Training a low-capacity $f_{\theta_{teacher}}$ on the backdoor-poisoned $D_{train}$ for a few epochs. (2) Weighting each data in $D_{train}$ according to its loss on $f_{\theta_{teacher}}$, and then training $f_{\theta_{student}}$ on the weighted $D_{train}$.**

data result in lower absolute values of their negative weights. Consequently, the impact on the clean accuracy of $f_{\theta_{student}}$ on clean data is limited.

## 5 EXPERIMENT

**Attack Setup** We evaluated our LfD defense method against seven dirty-label backdoor attacks and two clean-label backdoor attacks on CIFAR-10[25], GTSRB[45], and an ImageNet subset[7]. Specifically, dirty-label attacks include BadNets with white square pattern (referred to as BN-W)[13], BadNets with grid square pattern (referred to as BN-G)[13], Trojan attack with square pattern (referred to as TJ-SQ)[35], Trojan attack with watermark pattern (referred to as TJ-WM)[35], $l2$-Invisible attack (referred to as L2)[28], Dynamic attack [38], Blend attack [6], and clean-label attacks include Sinusoidal signal attack (referred to as SIG) [1] and Clean-Label attack (referred to as CL) [50]. The target class for all backdoor attacks on CIFAR-10 and GTSRB datasets was uniformly designated as '1', whereas on the ImageNet-subset dataset, the target class for all backdoor attacks was consistently set to '0'. To ensure the success rate of CL and Signal attacks on the GTSRB dataset, we increased the injection rate to 9.5% instead of 8%. The specifics of the backdoor triggers are summarized in Table 1. We utilized the Adam optimizer with an initial learning rate of 0.001, zero weight decay, and Random Crop (padding = 4) with Random Horizontal Flip for data augmentation when training models on the poisoned trainsets. The CosineAnnealing learning rate decay scheduler was applied for 80 training epochs.

**Defense Setup** On the CIFAR-10 and ImageNet-subset dataset, we set $\alpha = 0.15$ and $\beta = 1e-3$, while on the GTSRB dataset, we set $\alpha = 0.2$ and $\beta = 3e-3$. We employed ResNet-S as $f_{\theta_{teacher}}$ across all

three datasets and utilized ResNet-18 as $f_{\theta_{student}}$ on CIFAR-10 and GTSRB, while on the ImageNet-subset, we employed ResNet-34 as $f_{\theta_{student}}$. During the training of $f_{\theta_{teacher}}$, no data augmentation is applied. For the training of $f_{\theta_{student}}$, we employ random cropping and random horizontal flipping as data augmentation. We compare LfD with three state-of-the-art defense methods: Fine-pruning (FP)[33], Activation-Cluster (AC)[2], and Anti-Backdoor Learning (ABL)[31]. For FP, AC, and ABL, we follow the configurations specified in their original papers, including the use of available clean data for fine-tuning/repair/training settings. For the ABL defense, we first trained the $f_{\theta_{teacher}}$ model for 20 epochs with a learning rate of 0.1 on CIFAR-10, ImageNet-subset and 0.01 on GTSRB, before reaching the turning epoch. Once we identified 1% of potential backdoor examples, we continued training the $f_{\theta_{student}}$ model for an additional 60 epochs on the entire training dataset, which helped to recover the model's classification accuracy. During the final 20 epochs, we trained the $f_{\theta_{student}}$ model using the LGGA loss with the 1% isolated backdoor examples and a learning rate of 0.0001. As for the Fine-pruning (FP) defense, we pruned the last convolutional layer of the $f_{\theta_{student}}$ model until its CA became similar to that of the other defense baselines.We utilized the activation clustering defense method from Trojan-zoo's open-source code. Firstly, we trained $f_{\theta_{teacher}}$ for 80 epochs without data augmentation on the ResNet-18 architecture with the poisoned trainset. Then, we extracted the activation values of all data in layer 4 of the training dataset and applied the K-means clustering algorithm to cluster them into two clusters. Finally, we trained a new model from scratch using the cluster with more data. All defense methods were trained with Random Crop (padding = 4) and Random Horizontal Flip.

**Table 1: Training setup for six backdoor attacks.**

| Attacks | Trigger Type | Trigger Patten | Inject Ratio |
|---------|--------------|----------------|--------------|
| BN | Fixed | Grid | 10% |
| Blend | Fixed | Random Pixel | 10% |
| TJ | Fixed | Reversed Watermark | 10% |
| Dynamic | Varied | Mask Generator | 10% |
| L2 | Fixed | Regularized Gaussian Noise | 10% |
| CL | Fixed | Grid and PGD Noise | 8%(other datasets) 9.5%(GTSRB) |
| SIG | Fixed | Sinusoidal Signal | 8%(CIFAR-10) 9.5%(GTSRB) |

**Metrics** We employ two metrics to evaluate the effectiveness of LfD. The first one is Clean Accuracy (CA), which measures the accuracy of $f_{\theta_{student}}$ in classifying clean test data. The second metric is Attack Success Rate (ASR), which quantifies the probability of $f_{\theta_{student}}$ classifying poisoned test data as the target label.

## 5.1 Effectiveness of Our LfD Defense

**Results on CIFAR-10** Table 2 presents the defense performances of 4 backdoor defense methods against the 9 backdoor attacks on CIFAR-10. Evidently, our LfD demonstrates optimal performance in reducing ASR while maintaining a high CA in most backdoor attacks. Compared to the best-performing defense method ABL, our LfD reduces the average ASR by 7.53% (0.67% vs. 8.20%) while outperforms ABL in terms of average CA by 5.56% (92.09% vs. 86.53%), respectively. This advantage becomes more pronounced compared to other defense methods.

The training accuracy on clean datasets without attack are also important, as significant degradation of accuracy suggests difficulties in correctly working in a normal scenarios without any attack, rendering the method impractical. Our method's accuracy on clean datasets outperforms ABL by 18.41% (91.68% vs. 73.27%). It is 0.77% less than AC (91.68% vs. 92.45%), However, AC has almost no defense effect when the poisoning rate is around 10%.

**Results on GTSRB** The results on GTSRB dataset are also presented in Table 2. It is evident that our LfD outperform other methods in most backdoor attacks. LfD reduces the average ASR by 5.18% (6.14% vs. 11.32%), 87.19% (6.14% vs. 93.33%), and 82.15% (6.14% vs. 88.29%) compared to ABL, AC, and Fine-prune.

We note that our LfD is not always the best when considering each attack individually. For instance, ABL exhibits the best defense against TJ-WM and Dynamic attacks on GTSRB. We suspect that this is due to the similarity between the triggers used in these two attacks and the feature of normal images. We suspect this is because even clean data may contain patterns similar to the trigger, making the detection of poisoned data more challenging [58]. If LfD fails to identify the majority of poisoned data, its defense effectiveness will be weakened. This limitation represents one of the shortcomings of our LfD approach and requires further improvement in future work.

**Results on ImageNet subset** As presented in Table 2. LfD achieves a higher CA and a lower ASR than other defense methods on all attacks. The accuracy of LfD is surprisingly higher than the result trained on a clean dataset without any defense (94.69% vs. 91.96%). We assume it is our dynamic discrimination strategy that makes

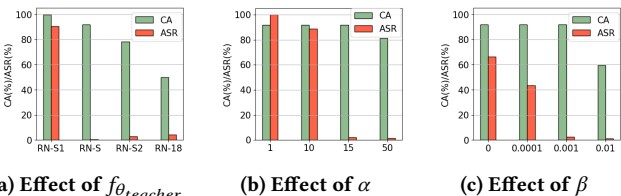

(a) Effect of $f_{\theta_{teacher}}$  (b) Effect of $\alpha$  (c) Effect of $\beta$

**Figure 4: The influence of the capacity of $f_{\theta_{teacher}}$, along with the hyperparameters $\alpha$ and $\beta$, on the CA and ASR of $f_{\theta_{student}}$.**

the model focus on learning difficult samples, resulting in better performance in a relatively complex dataset.

In summary, our LfD achieves superior performance against a wide range of attacks compared to other methods. This is due to LfD's ability to expose and identify backdoor samples via a low-capacity model, while utilizing dynamic discrimination strategy to learn and adapt to clean samples.

## 5.2 Ablation Studies

To gain a deeper understanding of LfD, we conducted a series of ablation experiments to elucidate the impact of hyperparameters, capacity of $f_{teacher}$, gradient ascent training, and dynamic discrimination method on LfD.

**Effect of $f_{\theta_{teacher}}$'s capacity** The capacity of $f_{\theta_{teacher}}$ influences the complexity of features it is capable of learning. We present in Figure 4a the CA and ASR of $f_{\theta_{student}}$ when utilizing four distinct models as $f_{\theta_{teacher}}$. The models employed are all derived from ResNet architecture,they are ResNet-S1(RN-S1)<ResNet-S(RN-S)<ResNet-S2(RN-S2)<ResNet-18(RN-18), and their detailed structures are provided in Section 5.3. The results indicate that when the capacity of $f_{\theta_{teacher}}$ is high, its capacity to learn complex features is enhanced. This results in a significant number of clean data instances having losses similar to those of poisoned data, subsequently prompting LfD to engage in gradient ascent training on a large number of clean data. Consequently, the CA of $f_{\theta_{student}}$ diminishes. Conversely, when $f_{\theta_{teacher}}$ has a excessively low capacity, its aptitude to learn features becomes exceedingly feeble, rendering it challenging to learn clean data and the majority of poisoned data. This results in the majority of clean and poisoned data have high losses. As a result, many poisoned data is either assigned positive weights or weights close to zero. Hence, the CA and ASR of $f_{\theta_{student}}$ remain largely unaffected.

**Effect of Hyperparameter $\alpha$** The variable $\alpha$ controls the amount of data used in gradient ascent training. We present the performance

**Table 2: Defense effectiveness of various defense methods on different backdoor attack methods. In the table, "No Defense" means no attack is applied, and "None" means no defense is applied.**

| Dataset | Attack Type | No Defense | | Fine-prune | | AC | | ABL | | LfD | |
|---------|-------------|------|------|------|------|------|------|------|------|------|------|
| | | CA | ASR | CA | ASR | CA | ASR | CA | ASR | CA | ASR |
| CIFAR-10 | BN-G | 91.51% | 100.00% | 85.22% | 99.98% | 91.17% | 100.00% | 91.49% | 0.51% | **92.62%** | **0.18%** |
| | BN-W | 91.89% | 99.95% | 82.67% | 98.71% | 91.29% | 92.57% | 89.98% | 3.26% | **92.01%** | **0.45%** |
| | Blend | 92.01% | 99.98% | 83.12% | 85.62% | 91.44% | 99.85% | 77.57% | 44.18% | **92.43%** | **3.38%** |
| | TJ-SQ | 92.95% | 100.00% | 83.90% | 66.87% | 94.24% | 99.96% | 92.09% | 0.36% | **92.17%** | **0.27%** |
| | TJ-WM | 92.89% | 100.00% | 85.42% | 61.02% | 92.38% | 99.32% | 90.18% | 0.04% | **91.76%** | **0.02%** |
| | Dynamic | 92.70% | 99.95% | 90.49% | 87.18% | 83.32% | 100.00% | 90.71% | 7.94% | **91.14%** | **0.55%** |
| | CL | 92.73% | 94.92% | 84.09% | 54.95% | 92.17% | 0.56% | 83.42% | 0.70% | **92.42%** | **0.64%** |
| | SIG | 93.15% | 92.16% | 84.76% | 76.32% | 65.77% | 87.37% | 89.48% | 6.12% | **92.41%** | **0.40%** |
| | L2 | 92.98% | 99.96% | 81.53% | 80.57% | 90.48% | 97.91% | 73.92% | 10.71% | **91.88%** | **0.11%** |
| | None | 92.64% | 0.00% | 88.56% | 0.00% | **92.45%** | **0.00%** | 73.27% | 0.00% | 91.68% | 0.00% |
| GTSRB | BN-G | 96.13% | 100.00% | 88.53% | 99.57% | 96.41% | 100.00% | 86.44% | 4.02% | **95.22%** | **0.02%** |
| | Blend | 95.42% | 100.00% | 85.90% | 99.50% | 96.28% | 80.84% | 82.09% | 11.36% | **92.31%** | **3.86%** |
| | TJ-SQ | 95.42% | 99.95% | 84.21% | 93.54% | 96.28% | 99.91% | **96.25%** | **16.06%** | 93.84% | 25.70% |
| | Dynamic | 96.35% | 99.92% | 88.38% | 99.84% | 95.96% | 99.96% | **96.67%** | **5.57%** | 88.39% | 2.76% |
| | CL | 95.88% | 43.85% | 89.71% | 59.26% | 87.06% | 83.45% | 81.82% | 26.29% | **95.62%** | **0.02%** |
| | SIG | 95.85% | 67.87% | 90.66% | 78.07% | 79.60% | 95.79% | 88.54% | 4.63% | **95.17%** | **4.45%** |
| | None | 96.90% | 0.00% | 90.14% | 0.00% | 90.46% | 0.00% | 83.95% | 0.00% | **93.53%** | **0.00%** |
| ImageNet subset | BN-G | 88.76% | 100.00% | 82.13% | 43.07% | 89.09% | 99.83% | 89.55% | 3.57% | **91.19%** | **0.31%** |
| | BN-W | 89.13% | 88.71% | 83.49% | 48.15% | 87.66% | 79.01% | 85.73% | 31.29% | **89.09%** | **3.01%** |
| | Blend | 91.54% | 99.69% | 84.79% | 98.77% | 90.80% | 17.59% | 84.44% | 79.16% | **89.83%** | **2.17%** |
| | CL | 92.41% | 79.76% | 82.30% | 59.72% | 87.94% | 80.59% | 83.06% | 36.05% | **90.07%** | **0.17%** |
| | None | 91.96% | 0.00% | 86.36% | 0.00% | 87.41% | 0.00% | 87.52% | 0.00% | **94.69%** | **0.00%** |

of $f_{\theta_{student}}$ under four different settings in Figure 4b, where $\alpha$ takes values of 1, 10, 15, and 50, respectively. The results demonstrate that a higher $\alpha$ leads to a better defense against backdoor attacks in LfD, as more poisoned data are isolated during training. However, we observed that increasing $\alpha$ also leads to a loss in CA, as more clean data in the gradient ascent training process can affect the model's performance.

**Effect of Hyperparameter $\beta$** The variable $\beta$ controls the importance of gradient ascent training relative to gradient descent training. We present the performance of $f_{\theta_{student}}$ under four different settings in Figure 4c, where $\beta$ takes values of 0, 1e-4, 1e-3, and 1e-2. The results demonstrate that a higher $\beta$ leads to a lower ASR as the intensity of gradient ascent training on poisoned data increases. However, we note that a high $\beta$ may also result in a loss of CA, as gradient ascent training on some clean data is also intensified.

**Gradient ascent training and dynamic discrimination method** To aid in understanding the impact of gradient ascent training and dynamic discrimination methods in LfD, we present in Figure 5 the CA and ASR of $f_{\theta_{student}}$ trained on three $D_{train}$ poisoned by BN-G, Blend and SIG separately, under three different scenarios with $\alpha$ set to 0.15: (a) training without gradient ascent, (b) training with gradient ascent while not utilizing dynamic discrimination strategy, and (c) training with gradient ascent while utilizing dynamic discrimination strategy for data weighting. Evidently, applying the gradient ascent training method can effectively reduce the ASR of $f_{\theta_{student}}$, however, training with gradient ascent on isolated data

indiscriminately can also lead to a decrease in CA. Compared to the solely unweighted gradient ascent training, utilizing dynamic discrimination strategy can lead to an enhancement in CA. This indicates that the gradient ascent training method can effectively suppress the insertion of backdoors, while the dynamic discrimination method can reduce the damage to the CA of $f_{\theta_{student}}$.

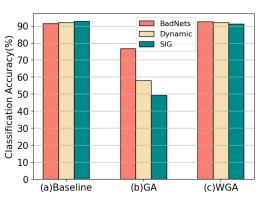

(a) CA of $f_{\theta_{student}}$

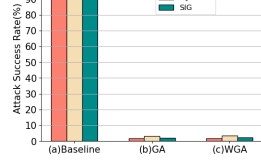

(b) ASR of $f_{\theta_{student}}$

**Figure 5: CA and ASR of $f_{\theta_{student}}$ trained on three $D_{train}$ poisoned by BN-G, Blend and SIG separately under: (a) training without gradient ascent, (b) training with gradient ascent cf. Eq. 5 (GA), and (c) training with weighted gradient ascent cf. Eq. 6 (WGA).**

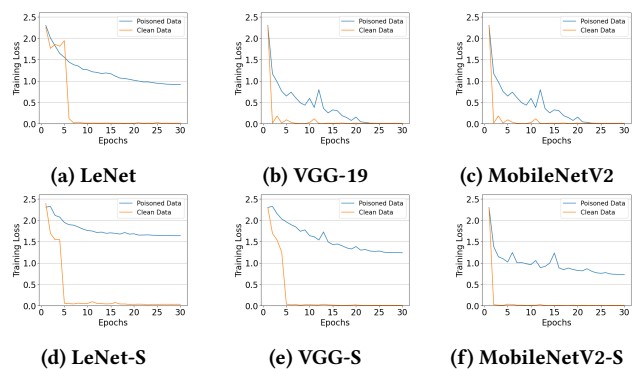

**(a) LeNet**  **(b) VGG-19**  **(c) MobileNetV2**

**(d) LeNet-S**  **(e) VGG-S**  **(f) MobileNetV2-S**

**Figure 6: The training loss of clean data and poisoned data on multiple model architectures, where the poisoned data are generated by BadNets backdoor attack and the models are constructed based on VGG, LeNet and MobileNet, their detiled informations are presented in Section 5.3. The experiments were conducted on the CIFAR-10 dataset with a poisoning rate of** 10%.

### 5.3 Exploration Across Various Model Architectures

We constructed low-capacity models on several other model architectures for additional exploration. We based our modifications on the ResNet architecture, altering the number of neurons and blocks to create ResNet-S, ResNet-S1, and ResNet-S2. For ResNet-S, ResNet-S1, and ResNet-S2, the first four layers consist of only one block each. In ResNet-S, these blocks contain 2, 2, 2, and 4 neurons respectively. For ResNet-S1, each block in the first four layers contains 1 neuron. In the case of ResNet-S2, the number of neurons in the blocks of the first four layers are 64, 128, 256, and 512 respectively.

For the VGG [44] network, we modified the number of neurons in each convolutional layer to obtain the VGG-S network. Specifically, in the VGG-S network, the number of neurons in each convolutional layer are set as 4, 8, 16, 16, 32, 32, 32, and 512, respectively.

We reduced the number of neurons and fully connected layers to obtain LeNet-S [26]. Specifically, LeNet-S consists of 2 convolutional layers and 2 fully connected layers, with 1, 2, 20, and 10 neurons in each layer.

Finally, we made modifications to the MobileNetV2 [42] network and obtained the MobileNetV2-S network. MobileNetV2-S consists of 2 bottlenecks, where bottleneck 1 contains 1 block with 3 convolutional layers, and the number of neurons in each layer is 4, 4, and 8 respectively. Bottleneck 2 contains 2 blocks, each with 3 convolutional layers, and the number of neurons in the two blocks' convolutional layers are 4, 4, 8 and 8, 8, 8 respectively. In addition, there are 2 convolutional layers outside the bottleneck, where the number of neurons is 4 and 16 respectively. All these models are trained on CIFAR-10 dataset poisoned by the same attack strategy to ensure a consistent comparison.

As shown in Figure 6, the results indicate that the distinction in average loss between poisoned and clean data is consistently greater in low-capacity models compared to high-capacity models.

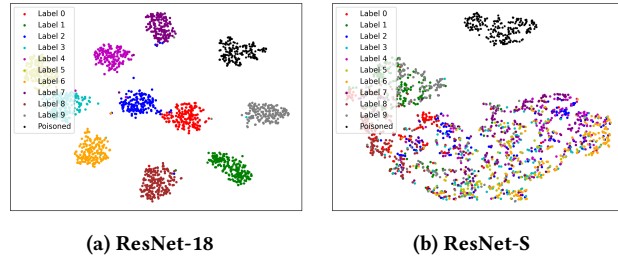

**(a) ResNet-18**  **(b) ResNet-S**

**Figure 7: The t-SNE visualizations of BadNets attack on ResNet-18 and ResNet-S models trained on CIFAR-10. Clean samples are distinguished by color based on their labels, while poisoned samples are marked in black.**

This suggests that this phenomenon is prevalent across various model architectures.

### 5.4 Visualization Analysis

To further understand how LfD works through a low-capacity model, we show the t-SNE [52] plots of the CIFAR-10 datasets (poisoned by BadNets attack) trained on both ResNet-18 and ResNet-S at twentieth epoch.

It can be clearly seen from Figure 7 that on the t-SNE plot of ResNet-18, the clean samples of each label and backdoor are clearly clustered and distinguished, while on the t-SNE plot of ResNet-S, only the backdoor samples are distinctively clustered, while the clean samples are still relatively chaotic. This indicates that in the early stages of training, ResNet-S performs well in backdoor feature learning, but poorly in semantic feature learning for the original classification task, while ResNet-18 performs well in learning both types of feature. Since our target is to separate the backdoor samples through the difference in loss, visualized results once again confirms that using a low ability model to distinguish backdoor samples is better than using a normal model.

It is a simple and effective method to reduce the learning ability of a CNN model by reducing the number of neurons and blocks. In our future work, we will (1) design new strategies to effectively reduce the learning ability of Transformer-style models to apply LfD in tasks of other modalities. (2) Develop appropriate date preprocessing methods to accentuate backdoor features to face possible adaptive attacks (which may intend to minimal loss differences to bypass our method).

## 6 CONCLUSION

In this work, we show the impact of model capacity on the distinction of the loss value between backdoored and clean data during training can be utilized to identify the poisoned data from a poisoned dataset. Based on this observation, we proposed a training-time backdoor defense method called Learning from Distinction (LfD), which employs a low-capacity teacher model to guide the training process of the student model by dynamically weighting the training data and thereby obtaining a clean student model. With extensive experiments, LfD demonstrated excellent performance in the robust training of neural networks against 9 state-of-the-art backdoor attacks.

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
