# OpenReview forum: "Learning from Distinction: Mitigating backdoors using a low-capacity model"
_acmmm.org/ACMMM/2024/Conference — MM2024 Poster_

### Official Review · Reviewer_iuS5 · 2024-05-19

**Rating:** 3
**Confidence:** 3

**Summary:**

The paper starts their work from the observation that poisoned samples have higher loss comparing to the clean samples and this phenomenon is more obvious in low-capacity models. Sufficient exploration on different model architectures and different datasets have verified this point. Based on these, Learning From Distinction(LFD) exploits a low-capacity teacher model to train a clean student model on backdoored datasets. Sufficient experiments on different datasets against 9 attacks show the method's effectiveness. Ablation study on hyper-parameters is enough to verify the method's correctness.

**Strengths:**

1. The observation of the huge loss descrepacy on low-capacity models is novel and has good contribution to the whole paper.
2. Extensive Experiments on various datasets shows good effect on defense.
3. The paper is well-organized and easy to follow.
4. The weighted method successfully utilizes the teacher model to supervise the student model to train a backdoor-free model.

**Limitations:**

Limitations:
1. The better performance of the benign accuracy comparing to the normal train on Imagenet should also be discussed.
2. High ASR on GTSRB against TJ-SQ is needed to be analyzed.
3. The construction of low-capacity model seems have no unified standards. How do the authors select the number of neurons in each layer?
4. The differences of the low-capacity model and the standard model is not clear enough. The comparison of the amount of parameters are better for the readers. Besides, the amount of the low-capacity models' parameters is also an important part needed to be discussed which can greatly enrich the whole paper.
5. Only ResNet models are explored in the main experiments, it would be better to show the vgg's results in the main experiments tables.

Suggestions:
1. It would be better to add the following citations:"Backdoor Defense via Adaptively Splitting Poisoned Dataset" "Backdoor Defense via Decoupling the Training Process" "Backdoor Defense via Suppressing Model Shortcuts"

2. It would be better to test the LFD on ISSBA.
"Invisible Backdoor Attack with Sample-Specific Triggers"

**Suitability:**

2

---

### Official Review · Reviewer_19G5 · 2024-05-21

**Rating:** 4
**Confidence:** 3

**Summary:**

This paper studies the backdoor defense of deep neural networks, proposing a training-time backdoor defense method based on distinguishing the loss value between backdoored and clean data during training. The method consists of two stages. Firstly, it trains a low-capacity model and utilizes it as a teacher model to supervise the backdoor-free student model. In the second stage, a dynamic weighting strategy is employed to select poisoned data from the poisoned datasets. Evaluations conducted on CIFAR-10, GTSRB, and ImageNet-subset datasets demonstrate the effectiveness of the proposed method.

**Strengths:**

- The paper is overall well structured. In particular, the preliminary analysis and the method part are logically coherent and structured clearly.
- The proposed approach is interesting and is effective against 9 well-known backdoor attacks.
- This paper reveals a highly important observation in backdoor attacks.

**Limitations:**

- Figure 4 demonstrates that three hyperparameters (f_{teacher}, \alpha, \beta) are highly sensitive, which may limit the practical applicability of this method. In practical scenarios, how can a defender select these three hyperparameters without knowing the attack information?
- The baseline choices for the evaluation are not up to date. The authors have overlooked some recently published backdoor defenses (e.g., [1-2]).
- (Not limitations) I'm curious about the relationship between teacher models and student models. Specifically, are there any interesting connections in terms of parameters and architecture? Additionally, what experimental conclusions can be drawn if the pre-training dataset is very large?

[1] Wang, Liu, et al. "Backdoor Defense via Deconfounded Representation Learning" CVPR 2023.

[2] Huang, Li, et al. "Backdoor defense via decoupling the training process" ICLR 2022.

**Suitability:**

2

---

### Official Review · Reviewer_bJzu · 2024-05-25

**Rating:** 4
**Confidence:** 4

**Summary:**

The paper proposes a new training-time backdoor defense method called Learning from Distinction (LfD), to counter backdoor attacks in neural networks. While traditional backdoor defense methods focus on detection and removal, LfD aims to prevent the learning of backdoor features during model training. The paper observed the impact of model capacity on the distinction of loss values between backdoored and clean data during training. Based on these observations, LfD utilizes a low-capacity model as a teacher to guide the learning of a backdoor-free student model through dynamic weighting. Experiments on various datasets demonstrate that LfD significantly reduces attack success rates while minimally affecting clean accuracy.

**Strengths:**

1. The paper theoretically analyzes the reason why low-capacity models can act as the teacher to help with backdoor defenses.

2. This paper conducts sufficient ablation studies to verify the effectiveness of each stage and each hyperparameter in the method.

3. The paper is well-written and easy to follow.

**Limitations:**

1. The threat model of the paper is not clear. LfD needs to control the training process of the model, however, some backdoor attack methods also need to control the training process of the model, such as Dynamic. How does the LfD defend against the attacks in this situation?

2. This paper only analyzed three kinds of backdoor attacks in the preliminary analysis, which cannot reflect the general characteristics of existing SOTA backdoor attacks.

3. The paper lacks comparisons against the SOTA backdoor attacks in the experiments, such as  WaNet [1], ISSBA [2] and BPP [3].

[1] Nguyen et al. ‘‘Wanet-imperceptible warping-based backdoor attack.’’ ICLR 2021

[2] Li et al. ‘‘Invisible backdoor attack with sample-specific triggers.’’ ICCV 2021

[3] Wang et al. ‘‘Bppattack: Stealthy and efficient trojan attacks against deep neural networks via image quantization and contrastive adversarial learn.’’ CVPR 2022

**Suitability:**

2

---

### Official Review · Reviewer_jpzk · 2024-05-26

**Rating:** 3
**Confidence:** 3

**Summary:**

The paper introduces a novel training-time backdoor defense method called Learning from Distinction (LfD). It focuses on using low-capacity models and dynamic discriminative strategies to reduce backdoor attacks in deep neural networks (DNNs), which enhances the robustness of the model, its acceptance of malicious attacks and its ability to maintain a high accuracy rate when using clean data. Experimental results show that the method can effectively reduce the success rate of malicious attacks and improve the accuracy of clean data.

**Strengths:**

1. Backdoor defence methods used in this paper take a low-capacity model to guide the training of a backdoor-free student model. Unlike traditional backdoor defence methods, the security of the model can be further improved.
2. The paper conducts extensive experiments on CIFAR-10, GTSRB, and ImageNet-subset datasets to evaluate the performance of LfD against various backdoor attacks. The results show that the success rate of the attack is dramatically reduced and does not affect the accuracy of the plot data, and the whole evaluation process is more comprehensive.

**Limitations:**

1. This paper can be further improved by expanding the experimental section to include a wider range of datasets, especially those with different complexities and characteristics.
2. There are many state-of-the-art backdoor defence methods now. Please compare with more baselines to illustrate the superiority of the methods.
3. Even a clean model theoretically has a very small attack success rate. Why is it directly 0 in this paper?

**Suitability:**

2

---

### Meta-Review · Area_Chair_spFT · 2024-06-27

**Recommendation:** Accept (Poster)
**Confidence:** 4

**Metareview:**

The paper introduces a novel training-time backdoor defense method, guided by a low-capacity model.  Reviewers generally consider the method novel and effective and the experiments are comprehensive (to a large extent). Although some issues were raised, they were well addressed during rebuttal. All reviewers support acceptance.